# A Survey of the Magnetic Anisotropy Detection Technology of Ferromagnetic Materials Based on Magnetic Barkhausen Noise

**DOI:** 10.3390/s24237587

**Published:** 2024-11-27

**Authors:** Liting Wang, Changjie Xu, Libo Feng, Wenjie Wang

**Affiliations:** 1School of Mechanical Engineering, Inner Mongolia University of Science and Technology, Baotou 014010, China; 2023022215@stu.imust.edu.cn (L.F.); wwj@mail.nwpu.edu.cn (W.W.); 2School of Emergency Equipment, North China Institute of Science and Technology, Langfang 065201, China; xuchjvip@ncist.edu.cn

**Keywords:** magnetic Barkhausen noise, magnetic anisotropy, easy axis, stress, texture, detection parameterb

## Abstract

Magnetic Barkhausen noise (MBN) is one of the most effective methods for determining the easy axis of ferromagnetic materials and for evaluating texture and residual stress in a nondestructive manner. MBN signals from multiple angles and different magnetization sections can be used to characterize magnetic anisotropy caused by various magnetization mechanisms. This paper reviews the development and application of magnetic anisotropy detection technology, and the MBN anisotropy models that take into account domain wall motion and magnetic domain rotation are analyzed thoroughly. Subsequently, the MBN anisotropy detection devices and detection methods are discussed, and the application of magnetic anisotropy detection technology in stress measurement and texture evaluation is reviewed. From the perspective of improving detection accuracy, the influence of composite mechanisms on magnetic anisotropy is analyzed. Finally, the opportunities and challenges faced by current magnetic anisotropy detection technology are summarized. The relevant conclusions obtained in this paper can be used to guide the MBN evaluation of magnetic anisotropy in ferromagnetic materials.

## 1. Introduction

### 1.1. Easy Axis in Ferromagnetic Materials

The key components in mechanical equipment are mostly made of ferromagnetic materials, and the dynamic changes in their mechanical properties are crucial for determining the service life and reliability of each component. Typically, these key components often undergo stress and strain in multiple directions when operating in complex environments. At this time, ferromagnetic materials are prone to changes in mechanical properties, such as plastic deformation, hardness changes, and fatigue damage. Therefore, conducting mechanical property detection and evaluation on key components is an important way to improve the safety level of equipment and extend its service life. Among them, detecting the magnetic anisotropy of ferromagnetic materials is an important part of production safety, which helps to more comprehensively monitor the operating status of mechanical equipment.

Ferromagnetic materials are mostly polycrystalline and inherently possess magnetocrystalline anisotropy, which refers to the phenomenon where different crystal orientations inside the material exhibit varying magnetic properties. For example, when Fe single crystals are magnetized along the <100> crystal axis, the magnetization curve can easily reach saturation, whereas it is difficult to be magnetized to saturation along the <111> crystal axis [1]. This indicates that ferromagnetic single crystals exhibit magnetic anisotropy. The direction that is most easily magnetized is referred to as the direction of easy magnetization, and the corresponding crystal axis is called the easy axis.

Polycrystalline ferromagnetic materials are composed of many single crystals with different orientations, so the macroscopic magnetic anisotropy is determined by the orientation of the easy axes of all single crystals inside. In the unstressed and unmagnetized state, domain magnetization vectors generally align themselves along the cubic axis of the crystal, and the direction of easy magnetization is <100> [2]. Then, the macroscopic easy axis of the ferromagnetic material is the vector sum of the easy axes of individual domains. In polycrystalline materials, due to elastic tensile stress, the domain magnetization vector is rearranged in the <100> direction closest to the stress [3], thus changing the macroscopic magnetic anisotropy.

The slip surface generated during cold and hot processing of the material may cause processing-induced magnetic anisotropy [4,5,6,7]. Micro-residual stress and high dislocation density regions may occur in grains, thus modifying the direction and amplitude of the easy axis of the polycrystalline ferromagnetic material. Macroscopic nonuniform plastic deformation is also the induction source of magnetic anisotropy [8,9,10]. The micro-residual stress on the grains is also superimposed to form the larger macro-residual stress, which extends on many grains. The residual stresses, like the applied stress, affect the magnetic anisotropy of the material and the easy axis direction. In addition, some polycrystalline materials exhibit significant macroscopic magnetic anisotropy due to their strong texture, such as the Goss texture {110}<001> in oriented silicon steel, which results in inconsistent magnetic properties measured in different directions of the material [11,12,13]. The magnetic properties in specific directions are also related to various microstructures such as dislocations [14] and grain boundaries [15,16].

In summary, the crystallographic texture, processing technology, stress state and microstructure affect the magnetic anisotropy and easy axis of ferromagnetic materials. The macroscopic magnetic anisotropy of the final material is jointly determined by all the factors influencing the overall magnetic properties in different directions. The test results of magnetic anisotropy can indirectly reflect information such as the microstructure and stress state of the material.

### 1.2. Characterization of the Easy Axis Based on Magnetic Barkhausen Noise

Over the past four decades, magnetic anisotropy detection technology has garnered significant attention in the field of nondestructive testing. Compared to testing methods such as hysteresis loop and eddy current, magnetic Barkhausen noise detection technology has been widely used in magnetic anisotropy research due to its unique advantages, such as fast detection speed, significant signal characteristics, and simple detection methods.

MBN is a microscopic magnetic phenomenon. Ferromagnetic materials cause discontinuous magnetic changes under the action of time-varying magnetic fields. These magnetization jumps are caused by the discontinuous movement of the magnetic domain wall from one pinning site to another and the discontinuous rotation of the magnetic domain. When its amplitude is high enough, an induced voltage is generated in a nearby coil. The induced voltage is called magnetic Barkhausen noise. MBN is sensitive to microstructure parameters, including grain size [17,18,19], carbon content [20,21], grain boundary [15,16,22], pearlite and lamellar spacing [23], as well as stress and residual stress [24,25,26,27], and it also has high sensitivity. Since the grain size, dislocations, grain boundaries, precipitates, residual stress, etc. are randomly distributed inside the material, the distribution of the domain walls is also random. Under the action of a continuous external magnetic field, a large number of MBN signals with different strengths are generated. Therefore, the MBN anisotropy detection technology is used to study magnetic anisotropy and the easy axis of a ferromagnetic material. MBN signals are measured by the excitation magnetic field rotating around the test center with respect to a certain reference direction. The angular-dependent MBN technology has been successfully used to determine the direction and magnitude of the easy axis in pipeline steel, non-oriented and oriented silicon steel, ASTM 36 steel and other materials [28,29,30,31].

This paper introduces the factors affecting the magnetic anisotropy of ferromagnetic materials and elaborates on its relationship with the detection of MBN anisotropy. In Section 2, the theoretical models for magnetic anisotropy detection are reviewed, and the MBN evaluation principle of magnetic anisotropy is summarized. Section 3 discusses three types of MBN anisotropy detection devices and explores the influences of various detection parameters on magnetic anisotropy results. Section 4 reviews the applications of magnetic anisotropy detection technology in stress measurement and texture evaluation at home and abroad. In the final section, the development direction of magnetic anisotropy detection technology is proposed from a multidimensional perspective, promoting its application and promotion in engineering structures.

## 2. Detection Theories and Models

The two intrinsic mechanisms for ferromagnetic materials to generate MBN signals in a time-varying magnetic field are irreversible domain rotation and irreversible domain wall motion [32,33]. The former primarily occurs at higher applied magnetic fields away from the magnetic saturation state (magnetization section 1 shown in Figure 1). At this time, the discontinuous jump behavior of magnetic domains is weaker, resulting in relatively lower induced MBN signals. Irreversible domain rotation is primarily associated with the nucleation and growth of reverse domains at grain boundaries. Utilizing the distribution pole figure of MBN characteristic parameters within this section can effectively reflect the average magnetocrystalline anisotropy of polycrystalline materials.

The latter mainly occurs in the magnetization sections 2 and 3 depicted in Figure 1. The intensity of MBN jumps generated around the coercivity point is relatively high (magnetization section 2), inducing high-amplitude MBN signals. This section is primarily characterized by 180° domain wall motion, forming the main peak of the MBN envelope curve. Meanwhile, due to the fact that moving a 90° domain wall requires more energy, the smaller amplitude MBN signals that appears after the main peak (magnetization section 3) are associated with the 90° domain wall movement. Irreversible domain wall motion is primarily caused by the pinning effect of microscopic defects in materials, such as dislocations, voids, and grain boundaries, which are usually related to material processing technology. It is generally believed that the pole figure of MBN signal characteristic parameters extracted from magnetization sections 2 and 3 reflects the magnetic anisotropy caused by material processing.

When ferromagnetic materials are subjected to external loads, stress-induced magnetic anisotropy affects the entire MBN envelope curve, and together with other mechanisms, it determines the macroscopic magnetic anisotropy of the material.

### 2.1. MBN Anisotropic Model Considering Domain Wall Motion

From the above analysis, it can be seen that the intensity of MBN jumps generated by the irreversible domain wall motion around the coercivity of the material is large enough to overwhelm the smaller MBN jumps generated by the irreversible domain rotation. Therefore, the magnetization mechanism considered in early studies on MBN signals is the irreversible motion of the 180° domain wall. MBN signals are very similar to damped resonance, which can be modeled by domain wall velocity theory. Currently, the widely used MBN models include the Jiles–Atherton (J-A) model [34,35,36,37], the Alessandro, Beatrice, Bertotti, Montosori (ABBM) model [38,39], the Sakamoto model [40], and the Han–Hauser model [41,42,43,44].

The first investigation of domain wall velocity was carried out based on a simple physical model [45,46,47]. In the model, it is assumed that a domain wall in a single crystal moves along a known crystal orientation. The velocity of the domain wall is given by *v*∝*H_a_*-*H_c_*, where *H_a_* is the applied magnetic field and *H_c_* is the coercive force of the material. Based on this, the ABBM model employs stochastic methods to describe the stochastic characteristics of MBN signals. The dynamic equation describing the domain wall motion in ferromagnetic materials [48,49,50] can be expressed as
(1)$$\sigma G\left(d\phi /dT\right)=H_{1}-H_{c}$$
where *σ* is the electrical conductivity; G = 0.1356 is a dimensionless constant; *H*_1_ is the applied field component parallel to a particular domain wall; d*ϕ*/d*T* is the rate of change in magnetic flux. Formula (1) is a step equation. Unless the applied magnetic field component *H*_1_ oriented along the magnetic domain wall overcomes the coercive field, the domain wall will not start to move. This model assumed that MBN signals are primarily generated in the magnetization section 2 with a significant amount of domain wall motion surrounding the stronger regions of MBN jumps. This hypothesis has been verified in the measurement of the coercivity field distribution of non-oriented silicon steel.

The J-A model is a physical model established based on domain wall theory and the law of conservation of intrinsic energy. This model has been widely applied due to its few parameters, fast computation speed, and clear physical significance. With the continuous improvement and development of the model, various scholars have incorporated features such as frequency [51,52,53,54], temperature [55,56], dislocation density [57,58], and stress [59] into the J-A equation, leading to rapid advancements in hysteresis simulation. The J-A model is capable of describing the MBN signals of the entire hysteresis loop and can be utilized for fitting the experimental results of various materials.

To describe the magnetic anisotropy behavior of grain-oriented silicon steel, an energy-driven model considering magnetic microstructures such as magnetic domain structure and domain wall motion was proposed [60,61]. The Han–Hauser model aids in predicting the effects of various physical parameters, such as temperature, magnetostriction, and mechanical stress, on magnetic hysteresis phenomena. The Sakamoto model takes into account the intrinsic microstructure of materials. It correlates the MBN signal with grain size and obtains that the MBN jumps are inversely proportional to the square root of the grain size. The classic equations of the above several MBN models are all one-dimensional and cannot explain the magnetic anisotropy characteristics of ferromagnetic materials.

A research team of Queen’s University in Canada studied magnetic anisotropy in the 1990s and proposed the MBN anisotropy model [62,63,64,65,66]. If it is assumed that only one macroscopic easy axis exists in the polycrystalline material, the MBN signal is generated by two domains. The direction of one domain magnetic moment tends to be the macroscopic easy axis, and its generated MBN signals demonstrate an angular dependence. The direction of the other domain magnetic moment is randomly distributed, but its generated MBN signals demonstrate no angular dependence [62,63,64]. The final MBN energy model with angular dependence in pipeline steel is expressed as
(2)$$E_{energy}=\alpha \cos^{2}\left(\theta -\phi \right)+\beta$$
where *θ* is the angle between the applied magnetic field and the reference direction (e.g., rolling direction); *ϕ* is the angle of the easy axis relative to the reference direction; *α* represents the fitting parameter of the angular dependent components in the MBN signal and is related to the motion fraction of the 180° domain wall; *β* indicates angular-independent background noise in the MBN signal. The team of Queen’s University in Canada experimentally proved that the MBN anisotropy model established based on the consideration of the domain wall motion could be used to fit most of the experimental data obtained in the measurements on different detection surfaces under different excitation conditions. The MBN model showed the good applicability and high accuracy. This model has also been recognized by domestic and international research teams and is widely used to characterize the magnetic anisotropy of various ferromagnetic materials.

The detected surface of a material does not necessarily contain only one macroscopic easy axis. Taking the dual easy axes produced by the asymmetric processing technology of spiral welded pipes as an example, it is assumed that there are two easy axes with different directions inside the material, and each easy magnetization direction can be regarded as a magnetic dipole population. Therefore, the influences of the two magnetic dipole populations on the respective easy axis as well as the possible interactive dipole pair need to be considered. As a result, Equation (2) can be decomposed into multiple corresponding parts, and the quadrupole interaction term is introduced to represent the interacting dipole pairs. Since the model mainly focuses on the angular-dependent MBN energy change, the quadrupole term is simplified as the product of the cos^3^(*θ* − *ϕ*) contribution of each easy axis in the material and the constant I. The MBN anisotropy model for the final dual easy axis system is expressed as [67,68,69]
(3)$$E_{energy}=\alpha_{A}\cos^{2}\left(\theta -\phi_{A}\right)+\alpha_{B}\cos^{2}\left(\phi_{B}-\theta \right)+I\cos^{3}\left(\theta -\phi_{A}\right)\cos^{3}\left(\phi_{B}-\theta \right)+\beta$$
where *θ* is the angle between the applied magnetic field and the reference direction; *ϕ_A_* and *Φ_B_* are the directions of two different easy axes; *α_A_* and *α_B_*, respectively, represent the degree of anisotropy along the two different easy axes; and *β* indicates the background noise of the MBN signal aligned isotropically with respect to the easy axis.

The research team of Beijing University of Technology [70,71] has conducted extensive research on the evaluation of magnetic anisotropy in typical materials using MBN signals and proposed an MBN experimental evaluation method for magnetic anisotropy. An experimental system was constructed to test the MBN signals in different directions and their characteristic parameter pole figures in typical materials. The description model and representation method of magnetic anisotropy were studied. The empirical fitting model in the form of the third-order Fourier series obtained is more universally applicable to magnetization sections, MBN characteristic parameters, and material types. The specific equation is expressed as
(4)$$MBN_{x}=a_{0}+\sum_{n=1}^{3}\left[a_{n}\cos\left(n\omega t\right)+b_{n}\sin\left(n\omega t\right)\right]$$
where *a*, *b*, and *ω* are undetermined coefficients; and *t* is the angle of the external magnetic field relative to the reference direction.

### 2.2. MBN Anisotropy Model Considering Domain Rotation

In recent years, the vigorous development of microstructural observation techniques such as electron backscatter diffraction, atomic force microscopy, and scanning electron microscopy has provided new research directions for the study of magnetic anisotropy in ferromagnetic materials. Some research teams [72,73,74] have turned their attention to the smaller MBN jumps generated by domain rotation as the magnetization mechanism in magnetization section 1, primarily focusing on analyzing the magnetocrystalline anisotropy directly related to the crystallographic texture of the material.

The Goodenough model [75] is the first model of reverse domain nucleation and growth proposed by the Lincoln Laboratory of the Massachusetts Institute of Technology. In polycrystalline materials, the regions with different crystal orientations are separated by grain boundaries. When the strength of the magnetic field is low, the magnetization vectors of adjacent grains cannot rotate from their easy axis to the direction completely aligned with the magnetic field. Therefore, the magnetization vector component perpendicular to the grain boundary is generally discontinuous so that magnetic free poles appear at the grain boundary. The nucleation of reverse domains is related to the density of magnetic free poles at the grain boundaries. The magnetic free pole density at the *i*-th grain boundary is expressed as
(5)$$\omega_{i}^{*}=M_{s}\mu_{o}\left(\cos\theta_{1}-\cos\theta_{2}\right)$$
where *M_s_* is the saturation magnetization of the single crystal; *μ*_0_ is the permeability of the vacuum; *θ*_1_ and *θ*_2_ are the angles formed by the magnetization vectors of adjacent grains and the normal to grain boundary.

For the first time, Espina-Hernández et al. [76] demonstrated a strong correlation between the MBN signal in magnetization section 1 and magnetocrystalline energy (MCE). Based on this, Tu Le Manh et al. [73,77] combined the ABBM model, J-A hysteresis model, and energy-driven model to obtain the MBN energy model considering the material’s crystallographic texture, grain size, and carbon content, as shown in Equation (6).
(6)$$E_{MBN}\left(\eta |H_{\eta },p,d_{g}\right)\propto 2\rho \frac{J_{s}}{\mu_{0}}\bar{\chi_{d}}{\bar{d_{g}}}^{2}\left(H-H_{n}-H_{g}\right)\bar{\omega }\left(\eta |H_{\eta },p\right)$$
where *E_MBN_* is the energy of the MBN signal; *H_η_* is the applied magnetic field strength at each angular position; *p* is a function of the pearlite content; *d_g_* is the grain size; *ρ* is conductivity; *χ_d_* is differential magnetic susceptibility; *J_s_* = *μ*_0_*M_s_* is the saturation polarization; *H* is the applied magnetic field; and *H_n_* and *H_g_* are, respectively, the critical magnetic field strength for generating the reverse domain and the threshold magnetic field required for moving the domain wall after nucleation. This model describes the MBN signals generated by the nucleation and growth of reverse domains in magnetization section 1, and it can be used to explain the correlation between angular dependence MBN signals and MCE within this section.

Wang Liting et al. [78] conducted research on the theoretical model of MBN characterization for magnetocrystalline anisotropy. Based on the concept of coordinate transformation, a connection was established between the macroscopic reference coordinate system of the material and the microscopic crystal orientation. The microstructural parameters in the model were determined by metallographic microscopy and electron backscatter diffraction technology. The MCE of pipeline steel X60 and oriented silicon steel 30SQG120 as a function of magnetic field angle were simulated. An experimental system was constructed to test MBN signals from different angles, and the MBN envelope curve was obtained by band-pass filtering and moving average algorithm. The starting point A of the MBN envelope curve was determined by the threshold point of the MBN signal background noise, and the first intersection point of the 75% envelope peak and the MBN envelope curve was taken as the cutoff point B. Then, the root mean square (RMS) value in the A-B section was extracted as a characteristic parameter to characterize the magnetic anisotropy of the aforementioned materials. The experimental results obtained were in good agreement with the predictions of the MCE model, as shown in Figure 2. It is indicated that the MBN anisotropy detection technology can be used to evaluate the MCE of pipeline steel and oriented silicon steel.

### 2.3. MBN Anisotropy Model Considering the Domain Structure Changes Under Stress

The application of the stress onto a solid results in the change of the strain state, and then the corresponding magnetoelastic energy is generated. In addition, the strain also changes the magnetostrictive energy, thus affecting the magnetic domain structure inside the material. Taking the Fe single crystal as an example, since its magnetocrystalline anisotropy energy is several orders of magnitude larger than the magnetostrictive energy, the easy axis of the magnetic domain in the grain will be retained along the crystal axis direction before the external magnetic field reaches the saturation state. The applied stress only rearranges the magnetization vectors of the domains in the crystal axis direction which is the closest to the stress [79].

The magnetoelastic energy generated by the stress on the domain structure is expressed as
(7)$$E_{\sigma }=-\frac{3}{2}\lambda_{100}\sigma \cos^{2}\theta V$$
where *σ* is the applied stress; *λ*_100_ is the saturation magnetostrictive coefficient along the <100> direction; *θ* is the angle between the applied stress and the magnetization direction; and *V* is the volume of the domain. According to the principle of minimum energy, when *σ* > 0, the tensile stress makes the magnetization direction of the domain parallel to the direction of stress; when *σ* < 0, the compressive stress makes the magnetization direction of the domain perpendicular to the stress direction.

Bertotti et al. [80] regarded individual grains as a relatively independent magnetic interaction region. In other words, the domain configuration in the single crystal can be used to represent regions of locally interaction domains in the polycrystalline material. The scholars of Queen’s University in Canada [81,82] further found that the MBN anisotropy model established based on the consideration of the domain wall motion was still applicable to determine the direction of the easy axis under the applied stress. Based on this, Clapham et al. [81] conducted MBN anisotropy detection experiments under different stress directions and determined that the main mechanism for modifying MBN signals under stress was the change in the number of 180° domain walls in the local interaction region. Equation (8) has given the stress threshold *σ_T_* required for introducing or removing one 180° domain wall in the interaction region.
(8)$$\sigma_{T}=\frac{4\gamma_{180^{\circ }}an\left(n+1\right)}{3\lambda_{100}\left[b^{2}\cos2\theta -4\delta a\cos^{2}\theta n\left(n+1\right)\right]}$$
where *a*, *b*, and *T* are the height, width, and thickness of the crystal, respectively; *n* is the number of 180° domain walls; *δ* is the thickness of the 180° domain wall; *γ*_180°_ is the domain wall energy per unit area; and *θ* is the angle between the applied stress and the reference direction.

Wang Liting [83] extended the MBN model to assess magnetic anisotropy under stress conditions. Assuming that the crystal orientation of the material remains unchanged under stress, by considering the changes in the macroscopic magnetization direction of the material with and without stress, the angular-dependent MCE of pipeline steel X70 were obtained. The results are shown in Figure 3. It can be observed that stress alters the shape of the pole figure of MCE, which means it changes the positions of the hard and easy axes in the material.

## 3. Magnetic Anisotropy Detection Method

### 3.1. MBN Signal Detection Device

The MBN signal detection device is shown in Figure 4 and mainly composed of a signal excitation component, a signal acquisition component and a signal processing component. The signal excitation component includes a signal generator, a power amplifier, a yoke and an excitation coil and is mainly used to generate an alternating current signal by the computer-controlled signal generator. After being amplified by the power amplifier, it enters the excitation coil, forms a closed magnetic circuit in the yoke and the sample, and generates an alternating magnetic field. The changing magnetic field causes the discontinuous motion or rotation of the domains in the sample, which in turn generates the MBN signal. The signal acquisition component is composed of a detection coil and a Hall sensor, which can collect the magnetization information at 2 mm below the surface of the tested sample and convert it into a corresponding electrical signal. The signal processing component includes a filter, a data acquisition card, and a computer and is mainly used to amplify and filter corresponding electrical signals, which are collected with the data acquisition card and finally input into the computer for subsequent analysis and processing.

The MBN signals are often regarded as random signals, and their statistical characteristics are analyzed in the time domain (Figure 5a). In the process of material magnetization, with the surface tangential magnetic field strength as the abscissa and the MBN envelope obtained by the moving average as the ordinate, the butterfly MBN envelope curve is plotted (Figure 5b) to extract multiple characteristic parameters. Common characteristic values of MBN signals include root mean square (RMS) value, MBN envelope peak value *M_p_*, tangential field strength corresponding to the peak value *H_c_*, envelope half-peak 50%*M_p_* width, remanence *M_r_*, counts, etc. The RMS calculation formula is
(9)$$\mathrm{RMS}=\sqrt{\sum_{i}V_{i}^{2}/N}$$
where *V_i_* is the voltage signal amplitude; and *N* is the number of pulse voltages.

The MBN is a weak signal, and the original signal without amplification is at the microvolt level, which is very vulnerable to the influence of the external environment. Among these, temperature, electromagnetic interference, and mechanical vibration are the main external factors that affect the MBN detection results [84]. Temperature primarily affects the MBN signal in two ways. Firstly, temperature can cause thermal expansion and contraction in materials, resulting in thermal stress, which indirectly influences the MBN signal through stress. Secondly, temperature itself has an impact on the MBN signal. As the temperature increases, the MBN signal gradually decreases. When the material reaches its Curie temperature, its ferromagnetism will completely disappear. To achieve high-precision evaluation of the mechanical properties of ferromagnetic materials using MBN detection technology, the influence of temperature on the MBN signal must be taken into account. Wang et al. [85] established an MBN theoretical model for the two mechanisms of temperature effect and proposed a new method that quantitatively describes the correlation between temperature and MBN characteristic parameters. Guo et al. [86] studied the temperature effect on MBN stress detection and proposed an analytical model based on the average volume of MBN jumps, which provides guidance for the temperature compensation of detection results in engineering applications. When the operating voltage of the sensor is too high, the magnetic field strength used for specimen magnetization becomes stronger, leading to potential vibrations in the sensor and reducing its stability. Meanwhile, if the voltage is too low, the MBN signal will be too weak, making it susceptible to interference from other noise signals. Therefore, it is necessary to select appropriate excitation parameters. Furthermore, the MBN detection process is susceptible to interference from external magnetic fields, making it difficult to ensure the accuracy of the collected MBN signals. The interference from external magnetic fields can be addressed from two perspectives. One is to shield the interference source, while the other is to implement the corresponding electromagnetic shielding design for the detection components of the sensor. Copper foil material can be attached to different substrates for electromagnetic shielding.

### 3.2. MBN Anisotropy Detection Device

The orientation of the yoke relative to the sample surface determines the direction of the applied magnetic field. Therefore, the MBN signal can reflect the motion law of the internal domains of the material near the detection coil in the excitation magnetic field with a specific orientation. When the tested material is magnetically anisotropic, the MBN signal characteristic parameters measured in different directions on the surface of the tested sample are different. These characteristic values are plotted in the form of pole figures varying with magnetic field angle so that they can be used to analyze the magnetic anisotropy characteristics of the material.

#### 3.2.1. MBN Anisotropy Detection Device—Mechanical Rotation

The MBN signal anisotropy detection device was developed on the basis of the MBN signal detection device. The most common method is based on the traditional U-shaped yoke sensor. The sensor is mechanically rotated in a certain step *θ* to measure the MBN signals of different directions, or the test sample is fixed in a unidirectional direction (such as rolling direction) as a reference to conduct an angular-dependent MBN study by cutting the tested sample at different angles. The schematic diagram is shown in Figure 6. Capó-Sánchez et al. [28,31,63,87,88,89,90] determined the easy axis direction of ASTM 36 steel, pipeline steel, low-carbon steel, oriented silicon steel, and non-oriented silicon steel by using the MBN signal pole figure testing method, and they achieved the characterization of magnetic anisotropy for different ferromagnetic materials. The results indicate that the direction in which the MBN signal attains its maximum value is the direction of the material’s macroscopic easy axis.

The MBN signal is sensitive to the contact conditions between the sensor and the sample. In order to ensure the repeatable measurement conditions and improve the accuracy of angle control, Wang Xiuxiu of Beijing University of Technology designed a card slot with an equal angular spacing distribution [91] of 10° (Figure 7a). The physical picture after 3D printing is shown in Figure 7b. By mechanically rotating the MBN detection sensor and placing it in the corresponding card slot, the angular-dependent MBN signal detection was completed.

#### 3.2.2. MBN Anisotropy Detection Device—Automatic Rotation

The detection method of mechanical rotation has become one of the main methods for magnetic anisotropy research due to its simple operation, reliable results, and high cost-effectiveness. This experimental method involves mechanically changing the detection angle of the sensor, which requires a longer detection time and is prone to introducing errors such as lift-off, affecting the accuracy of the final detection results. To address the aforementioned limitations, scholars from various countries have introduced various automatic rotation methods to achieve the angular-dependent MBN signal detection.

Pérez-Benítez and Martínez-Ortiz et al. [6,92] built an MBN signal anisotropy detection device in the laboratory (Figure 8). The MBN detection in different directions was mainly achieved through a plane goniometer. The goniometer consisted of an acrylic base and a circular ruler. The base could rely on a stepper motor and a reduction gear mechanism to drive the sample table to rotate. The rotation step of the circular ruler was 1°. The stepper motor was driven by a function generator (Agilent 33220A) and could precisely control the rotation angle of the square sample. For the purpose of measuring the MBN signals of the sample in different directions, a probe was placed on the surface of the sample, and a lift-off device was added to vertically change the position of the probe before and after rotating the sample table. On this basis, Tu Le Manh et al. [93] developed a set of detection devices for the MBN anisotropy in circular samples. As shown in Figure 9, the device is composed of (a) an MBN signal acquisition system, (b) a circular goniometer, (c) a U-shaped yoke, (d) a tested sample, (e) magnetic pole pieces, (f) an angular scale, (g) a sample rotation knob, and (h) the vertical array of Hall sensors. Interestingly, in the device, the special magnetic pole piece and U-shaped yoke formed a closed magnetic circuit to magnetize the circular sample.

Caldas-Morgan et al. [94] used a device with permanent magnets to accurately control the rotating magnetic field. The method could realize the continuous excitation mode with two neodymium permanent magnets. The magnetic yoke, hall probe, and MBN coil were also packaged in a cylindrical case and mounted on a rotating device. Tens of thousands of MBN signals could be obtained through the continuous rotational Barkhausen method. The method could quickly and accurately determine the magnetic anisotropy of a material.

#### 3.2.3. MBN Anisotropy Detection Device—Multi-Pole Sensor

For the purposes of simplifying the operation steps and improving the measurement efficiency and accuracy, an MBN anisotropic detection device based on a multi-pole magnetic sensor has been developed. The control of the test angle is achieved by adjusting the input circuit parameters of each of the pole excitation coils. Yamada et al. [95] conducted a magnetic circuit analysis on a tetra-pole sensor and concluded that the magnetic permeability in various directions exhibited an elliptical distribution, and there was a linear relationship between the principal magnetic permeability and the principal stress. Vengrinovich et al. [96] adopted the tetra-pole sensor assembled from two mutually perpendicular U-shaped yokes. Each U-shaped yoke was separately excited, and the magnetic field was superimposed to achieve magnetic field excitation in any direction and obtain MBN signals. White [97] and McNairnay [98] improved the reactor branch pipeline with the tetra-pole sensor. Feedback systems were used for the magnetic flux control. The loading method with springs reduced the impact of the lift-off distance on the pipeline, thus allowing the measurements to be like that on a flat plate. This method could detect MBN signals in different directions on the curved surface. Leszek Piotrowski et al. [99] introduced a measuring device for rapidly determining the stress state of steel industrial components, which was also designed based on a tetra-pole sensor, and the direction of the magnetic field was synthesized by pairs of magnetic poles.

The Fraunhofer team in Germany [100] developed a special rotating magnetic field probe which was equipped with three magnetic poles separated by 120° to realize the angular-dependent MBN signal detection. Ran Deqiang [101], Wang Xiuxiu [91], and Jin Xiaokun [102] of Beijing University of Technology developed an MBN anisotropy automatic detection system equipped with a three-pole sensor (Figure 10). The magnetic field direction and the uniformity of the field distribution in the central area of the three-pole sensor were analyzed by simulation and experiments, which confirmed the feasibility of the rotating field excitation method. The four-channel function generator was controlled by LabVIEW software so that the three-way excitation signal output could synthesize a rotatable excitation magnetic field at the center of the three-pole sensor, achieving the characterization of residual stress and residual plastic strain based on the MBN anisotropy. In addition, Shi et al. [103] developed a six-pole sensor to detect plane stress. Isono et al. [104] developed a nine-pole sensor, with the central pole serving as the excitation pole, surrounded by an excitation coil, and the eight peripheral poles serving as the detection poles, which were surrounded by an induction coil. The biggest advantage of this design was that it eliminated the need to rotate the excitation magnetic field, allowing for the acquisition of mechanical property information in different directions of the material with a single measurement.

In summary, the above three types of MBN anisotropy detection devices have their own advantages and disadvantages, and they are commonly applied in different fields and under various detection conditions, as shown in Table 1. The detection method of mechanical rotation is easy to operate, reliable in results, and cost-effective, but it is greatly influenced by human factors. The automatic rotation detection method enhances detection efficiency and accuracy. However, two factors may interfere with obtaining accurate MBN signals: firstly, the rotational motor controlling the magnetic field rotation may generate electromagnetic interference, reducing the accuracy of the obtained MBN signal; secondly, the rotational speed of the sensor can affect the sensitivity of the measurement signal. The detection method of multi-pole sensors can be used for online and rapid detection, but such sensors are cumbersome to manufacture, costly, and require high uniformity of the magnetic field in the detection area.

### 3.3. Effects of Detection Parameters on Magnetic Anisotropy Results

MBN signals are closely related to measurement methods and detection parameters. Therefore, experimental factors such as the frequency, current, excitation period, and sensor structural dimension of the excitation magnetic field have significant effects on the magnetic anisotropy results of the material.

#### 3.3.1. Effects of the Excitation Magnetic Field Frequency on Magnetic Anisotropy Results

Due to the fact that changes in magnetic flux density always lag behind changes in magnetic field intensity, the amplitude of magnetic flux density gradually decreases from the surface of the material inward, which leads to the formation of the skin effect. The smaller the penetration depth, the smaller the range of material magnetization, the fewer the number of domain wall movements and magnetic domain rotations, and the weaker the resulting MBN signals. This determines that the sensitivity of the MBN signal to the magnetic characteristics of the material surface is higher than that to the deeper parts of the material. Therefore, it is impossible to obtain strong MBN signals from areas farther away from the surface. Generally speaking, the MBN signal can reflect changes in the microstructure and stress state within a depth range of approximately 2 mm from the surface of the material. The magnetization depth of the material can be determined according to Equation (10).
(10)$$\delta =\frac{1}{\sqrt{\pi \mu \sigma f}}$$
where *f* is the magnetic field frequency; *σ* is the conductivity; *μ* is the permeability; and *δ* is the penetration depth of the MBN signal. To obtain a strong MBN signal suitable for magnetic anisotropy research, a lower excitation magnetic field frequency (*f* < 100 Hz) should be selected.

Vashista et al. [105] studied the effects of high-frequency excitation (125 Hz) and low-frequency excitation (0.4 Hz) on MBN envelope. It revealed that the change in MBN envelope shape under low-frequency excitation could better reflect the magnetization process caused by different microstructural changes. Singh et al. [106] found that when the excitation magnetic field frequency varied within the range of 20~50 Hz, the root mean square value of MBN increased with the increase in excitation frequency. Stupakov et al. [107] studied the influences of excitation magnetic field frequency on the MBN envelope and spectrum by using a surrounding test coil and a coil placed on the surface of the test piece, respectively. Veronika Păltânea and Fiorillo et al. [88,108,109] conducted MBN detection experiments by cutting samples according to different angles with respect to the rolling direction, and the experimental data were analyzed with the interpolation method. Finally, it was determined that the easy axis of the material depended solely on the internal magnetic flux density and was independent of the excitation magnetic field frequency. Wang et al. [71] applied MBN technology to evaluate the magnetic anisotropy of pipeline steel and silicon steel, and they analyzed the influences of the frequency and current of the excitation magnetic field on the magnetic anisotropy results. By introducing a proportionality coefficient *k*, the degree of magnetic anisotropy of the materials was quantitatively described, and the optimal detection parameters of each material were determined.

#### 3.3.2. Effects of the Excitation Magnetic Field Current on Magnetic Anisotropy Results

The maximum applied magnetic field strength *H_amax_* of the sensor is determined by the current of the excitation magnetic field, the number of coil turns, the geometry of the yoke and the core material, and it can be expressed as
(11)$$H_{a\max}=\frac{Ni}{L}$$
where *N* is the number of excitation coil turns; *i* is the excitation magnetic field current; and *L* is the effective magnetic circuit length. Given a selected sensor, the maximum applied magnetic field strength mainly depends on the excitation magnetic field current.

The shape of the multi-angle MBN signal pole figure varies with the applied magnetic field strength. The results of numerous scientific research teams have shown [28,110,111,112,113] that the variations in MBN signals with the magnetic field angle under different applied magnetic field strengths could provide three different types of magnetization dynamics information, corresponding to the magnetization sections 1, 2, and 3 shown in Figure 1 (reverse domain nucleation and growth, 180° domain wall motion, and 90° domain wall motion). Martínez-Ortiz et al. [113] studied the influences of different applied magnetic field strengths on the easy axis of pipeline steel. Due to different excitation magnetic field currents involving various magnetization processes, the material exhibited distinct easy magnetization directions.

#### 3.3.3. Effects of the Excitation Period on Magnetic Anisotropy Results

The magnetization process of polycrystalline ferromagnetic materials is a complex phenomenon, but it is widely believed that the process can be divided into some major magnetization sections: reversed domain nucleation, 180° domain wall motion and 90° domain wall motion [64,92,114]. The MBN signals generated by 180° domain wall motion are influenced by the reverse domain nucleation process, while the remaining magnetization sections are relatively more independent. Pérez-Benítez et al. [92] proposed a new method for dividing the magnetization sections based on the test method of the MBN signal energy pole figure, which could effectively identify three different types of magnetization dynamics information. On this basis, Chávez-Gonzalez et al. [115,116] studied the effects of different magnetization periods on the results of magnetic anisotropy, and they pointed out that the differences in the shapes of the obtained pole figures were related to different magnetization dynamics processes. Due to the need to consider signal changes in both the time and frequency domains for magnetization dynamics information, traditional single time-domain or frequency-domain analysis methods can no longer meet the requirements of detection. Scholars from various countries have begun to introduce time-frequency analysis methods to process MBN signals. Maciusowicz et al. [117,118] proposed a method for converting MBN data based on the Short-Time Fourier Transform (STFT), and they applied it to divide the magnetization sections, achieving the evaluation of magnetic anisotropy in silicon steel materials.

#### 3.3.4. Effects of Sensor Structural Dimension on Magnetic Anisotropy Results

Given the material being tested, the structural dimensions of the sensor directly affect the strength of the MBN signal and the sensitivity of detection. The extracted characteristic parameters can reflect the microstructural information of the material being tested in the local area between the two magnetic poles of the sensor. Wang et al. [70] explored the influences of the magnetic circuit spans and detection positions of the sensor on the MBN evaluation results of the magnetic anisotropy of typical ferromagnetic materials. Figure 11 shows the RMS pole figure measured in pipeline steel X70 using three sensors with different magnetic circuit spans. The results indicated that the magnetic circuit span of the sensor had a significant impact on the magnetic anisotropy results, and sensors with a larger span were better able to reflect the intrinsic microstructural information of the tested material.

Overall, MBN signals are susceptible to various experimental factors, and detection parameters such as the frequency, current, excitation period, and sensor structural dimension of the excitation magnetic field can affect the detection results of material magnetic anisotropy. For materials under different conditions, the optimal detection parameters are different. Determining the optimal detection parameters is one of the key factors required in order to obtain the optimal MBN signal. Therefore, the developed MBN anisotropy detection technology scheme and the obtained relevant conclusions can be used for the MBN evaluation of magnetic anisotropy of ferromagnetic materials.

## 4. Magnetic Anisotropy Detection Technology and Application

MBN, as a nondestructive testing and evaluation technology, has garnered significant attention due to its rapid detection speed and reliable results. MBN signals are closely related to the intrinsic microstructure and structural factors of materials. MBN anisotropy detection technology can be used to study the magnetic anisotropy of materials. With the in-depth study of magnetic anisotropy detection theory and detection methods, the MBN anisotropy detection technology has been widely applied in the detection and evaluation of engineering structures.

### 4.1. Application in Stress Measurement

The application of the stress onto a solid results in the change in the strain state, and then dislocations, holes and cracks are generated inside the material. In addition, the variable stress also produces local stress concentration. On the one hand, the potential barriers generated by these defects hinder domain wall motion and magnetic domain rotation; on the other hand, stress also accelerates the fusion of magnetic domain structures with similar orientations [119]. From this perspective, the influences of stress on MBN signals are the result of the interaction between the two aforementioned effects. When the stress is lower than the yield strength, the material undergoes elastic deformation, and then fewer defects such as holes and cracks are generated inside the material. The stress mainly increases the MBN signal, and the measured MBN signal characteristic parameters have a good linear relationship with stress. As for the materials with a positive magnetostriction coefficient *λ* [79,120,121,122,123,124], increasing tensile stress increased the area of the 180° domain wall, leading to an increase in the amplitude of the MBN signal. Increasing the compressive stress reduced the area of the 180° domain wall, reducing the amplitude of the MBN signal. When the stress exceeds the yield point, the material enters the plastic deformation stage, and then residual stress is generated inside the material. Simultaneously, local pinning effects of magnetic domains in the form of dislocation, dislocation entanglement and cellular substructure are formed. This leads to a predominantly inhibitory effect of stress on the MBN signals, and a complex nonlinear mapping relationship exists between the MBN signal and stress [125,126,127,128]. Based on the above two aspects, the stress and strain of ferromagnetic materials can be obtained by analyzing the characteristic parameters of the MBN signal.

In practical applications of engineering structures, key components are often subjected to stress in multiple directions during operating in complex environments, exhibiting significant stress concentration. To describe the stress state of a point, apart from paying attention to the stress magnitude itself, it is more crucial to obtain the magnitude and direction of its principal stress. This implies that ferromagnetic materials exhibit strong magnetic anisotropy under stress conditions. Stress-induced magnetic anisotropy reflects the characteristics of the magnetization vector in the material related to the stress direction, and it influences the preferential orientation of magnetization through magnetoelastic energy. In practical engineering scenarios, how to truly achieve the measurement of one-point stress state is a pressing challenge that needs to be addressed in the engineering community. Grijalb et al. [129] found that the varying stress distribution on the specimen surface exhibited similar patterns to the distinct distribution of MBN signals. Krause et al. [81,82] experimentally observed that the shape of the pole figure formed by the MBN signal energy measured in multiple directions changed with the applied stress. By comparing the pole figures under stressed and unstressed conditions, the direction of stress could be determined. Capó-Sánchez et al. [31] utilized the MBN signal energy pole figure testing method to determine the easy axis direction of ASTM 36 steel, and they analyzed the effects of uniaxial stress on the direction of the easy axis. At the Beijing University of Technology, Ran Deqiang [101] used the strain rosette principle to predict the magnitude and direction of the principal stress in materials. Zheng Yang et al. [130] proposed a plane stress tensor measurement method based on the circumferential MBN distribution. The method realized the demodulation of normal stress and shear stress in any direction of the measurement point, and it could obtain the magnitude and direction of the maximum principal stress.

When residual stress exists inside the material, the residual stress affect the microstructure of the material, altering the magnetic properties of the material. Based on this, the distribution of residual stress inside ferromagnetic materials can be quantitatively analyzed through MBN signals. Currently, numerous research teams have conducted studies in this area. Schreiber and Vengrinovich et al. [131,132] extracted the root mean square as the eigenvalue to characterize residual stress by measuring the MBN signals in two mutually perpendicular directions, and they evaluated the residual stress state in the two-dimensional plane. In addition, the distribution of residual stress was intuitively displayed. Cikalova et al. [133] used an autocalibration method to analyze the two-dimensional stress state and established a fixed calibration function for predicting residual stress. This method realized the evaluation of the plane stress state. Pal’A and Stupakov et al. [134,135] adopted a loading–unloading experimental mode with low-carbon steel materials. The excitation was performed in the direction parallel and perpendicular to the tensile direction in order to measure MBN signals in the two directions. The microstructure under different strains was observed with a transmission electron microscope. The different changes in the two directions were ascribed to the dual effect of residual stress and dislocation density. Stefanita et al. [136] studied the effects of elastic and plastic strain on angular-dependent MBN energy pole figures and found that elastic strain could stimulate more significant new easy axes.

Based on the above analysis, it can be seen that the research on MBN in stress measurement mainly focuses on two major topics. The first major topic is the adoption of phenomenological methods to establish a quantitative magnetic prediction model for stress. This model can analyze the correlation between stress and MBN characteristic parameters in a specific direction of ferromagnetic materials and obtain the stress magnitude at a detection point along a one-dimensional angle. For engineering structures, to describe the stress state at a point in the material, it is necessary to obtain the stress tensor and determine the magnitude and direction of the principal stress. Therefore, the measurement direction at the same detection point plays a crucial role in influencing the MBN signal. The MBN anisotropy detection technology can be used to characterize the plane stress of ferromagnetic materials, which is also another topic based on the research of MBN technology in stress measurement.

In the actual stress detection process, the characteristic parameters detected by the MBN sensor are influenced not only by the material’s own properties but also by various factors such as the magnetic field strength, the geometry of the yoke, and the degree of coupling between the sensor and the tested component. Therefore, optimizing the sensor structure and excitation parameters has a crucial impact on improving the sensitivity and accuracy of stress detection. The influence of excitation parameters primarily stems from magnetic field strength and excitation frequency [137,138,139,140]. Selecting appropriate excitation parameters can enhance the sensitivity of stress detection. Santa-Aho et al. [141] utilized the MBN magnetization voltage sweep method to detect the case-hardened depth. Neslušan et al. [142] analyzed the influence of magnetization conditions on the characteristic parameters of MBN, and they determined the optimal excitation parameters through frequency and amplitude sweep experiments. Therefore, the experimental method of amplitude and frequency sweeping is an effective approach to optimize the MBN excitation parameters. On the other hand, optimizing the sensor structure is crucial for enhancing the sensitivity, repeatability, and accuracy of the sensor. Many scholars have adopted the finite element method to analyze the impact of structural parameters of sensors, such as the shape of the yoke [143], the number of turns of the excitation coil [144], the shape and number of turns of the detection coil [144], and the lift-off [145] on the MBN signal. This provides strong technical support for ensuring the stability and reliability of the signal during continuous measurement, which is also a technical difficulty that must be overcome during stress detection.

### 4.2. Application in Texture Evaluation

Magnetocrystalline anisotropy is an inherent property of materials, primarily describing the magnetic anisotropy of a single crystal, which is closely related to the magnetization direction relative to the crystal axis. In polycrystalline ferromagnetic materials, the orientations of most of the grains are randomly distributed, so the material behaves isotropically from the macro-perspective. When the orientations of the grains of the polycrystal are gathered together, the texture phenomenon appears in polycrystalline materials, thus causing anisotropy in the structure and properties of the material. The macroscopic magnetic anisotropy is affected by the magnetocrystalline anisotropy of individual crystal and the grain orientation distribution. In some special materials, the presence of texture helps to improve the mechanical properties of the material in specific directions, but in most materials, the emergence of texture should be avoided. Several major process stages in alloy production (hot rolling, normalizing, cold rolling, annealing, etc.) are all likely to produce texture. Therefore, the study of texture has always been a research hotspot in the engineering field.

X-ray diffraction is one of the main methods for evaluating the texture. This method can accurately provide the texture type, distribution and volume fraction of texture, but this method is destructive, and the detection process is cumbersome. In addition, the slow detection speed is not suitable for online detection. Therefore, Espina-Hernández et al. [76] combined MBN anisotropy detection technology with X-ray diffraction to study the texture of the material, solving the problem of being unable to evaluate texture online.

The number of grains in X-ray diffraction measurement is generally in the order of hundreds of thousands or even millions. In actual MBN anisotropy detection, the MBN signal detected with an induction coil only reflects the motion law of magnetic domains in a local area. To enhance the accuracy of nondestructive evaluation of texture, scholars have proposed utilizing electron backscatter diffraction (EBSD) as a means of texture detection. EBSD is a characterization technique of local texture and has been widely used to study the effects of grain size, grain orientation, grain boundary distribution, and magnetic free pole density on MBN signals in ferromagnetic materials [146,147,148]. Tu Le Manh et al. [72,149] estimated the average MCE of pipeline steel by measuring the set of individual grain orientations in the local area with the EBSD technology. The estimated averages were consistent with the MCE obtained by MBN measurements, indicating that the MBN technology could determine the MCE caused by the change in the crystallographic texture of the material. With grain-oriented silicon steel as the study object and the grain orientation as the only variable, Campos et al. [150] carried out angular-dependent MBN detection and found that irreversible domain rotation consumed more energy than DW motion. He et al. [151,152] measured the MBN signals and texture factors in different directions in non-oriented silicon steel, and they observed a strong correlation between the root mean square of the MBN signals and the texture factors. Therefore, the magnetic anisotropy of non-oriented silicon steel could be evaluated and the texture factor could be analyzed by measuring MBN signals in different directions.

Up to now, the MBN anisotropy detection technology has gradually become an important means for the nondestructive evaluation of texture, providing a solution for the online nondestructive detection of magnetic anisotropy in polycrystalline ferromagnetic materials. However, in practical engineering applications, the service environment of key components is complex and variable. The obtained magnetic anisotropy results are influenced by a combination of mechanisms such as average magnetocrystalline anisotropy and stress-induced magnetic anisotropy. It is not accurate enough to directly use the MBN anisotropy detection results to characterize stress or texture. Therefore, it is necessary to consider the influence of compound mechanisms on magnetic anisotropy and study the impact of magnetization dynamics on MBN stress detection and the nondestructive evaluation of texture results.

### 4.3. Effects of Compound Mechanisms on Magnetic Anisotropy

In existing studies on the magnetic anisotropy of materials, the strongest MBN signal was usually used to identify the macroscopic easy axis, but the source of the magnetic anisotropy was generally ignored. The magnetic anisotropy analyzed in most studies was related to stress or plastic deformation, and only the dominant role of stress-induced magnetic anisotropy in these studies was considered. The Mexican team was primarily dedicated to evaluating the magnetocrystalline anisotropy directly related to the crystallographic texture of materials. In addition, numerous scholars have studied the magnetic anisotropy induced by the rolling process. From this perspective, most scholars focused on one influence mechanism in their research on magnetic anisotropy, which was clearly not applicable to the context of engineering applications. In recent years, scholars have noticed this issue and begun to explore the effects of compound mechanisms on magnetic anisotropy and the easy axis. He et al. [151,152] explored the changes in the residual stresses and texture of non-oriented silicon steel in various stages of the heat treatment process. If only partial grains in the material were recrystallized, the superposition of residual stress and magnetocrystalline anisotropy caused MBN signals to deviate from the cosine curve. Both residual stress and magnetocrystalline anisotropy jointly determined the magnetic anisotropy of the material. Amiri et al. [153,154] studied the effects of the applied stress and residual stress on MBN signals and proved the competitive relationship between magnetocrystalline anisotropy and stress-induced magnetic anisotropy in determining the magnetic anisotropy of materials. Under the low applied stress, the magnetocrystalline anisotropy played the leading role, and the magnetic domains were converted into the easy axis of the crystal. Under the higher applied stress, the stress-induced magnetic anisotropy activated a new easy axis, forcing the domains to rotate toward the new easy magnetization direction controlled by the stress.

Generally speaking, the magnetic anisotropy of materials is primarily influenced by the combined effects of three mechanisms: magnetocrystalline anisotropy, processing-induced magnetic anisotropy, and stress-induced magnetic anisotropy. The magnetic anisotropy determined by MBN anisotropy detection depends on the main influence mechanism; thus, it is not possible to conduct a detailed analysis of all factors simultaneously that affect the overall magnetic anisotropy using this method. Recently, some scholars have introduced the concept of magnetization dynamics, utilizing time-frequency analysis methods to divide MBN signals into different magnetization sections. This approach is employed to characterize magnetocrystalline anisotropy, processing-induced magnetic anisotropy, and other related phenomena. These analytical methods are helpful to explain the sources of anisotropy in ferromagnetic materials, and they enhanced the accuracy of subsequent stress detection and texture evaluation. Currently, commonly used time-frequency analysis methods include STFT, Wavelet Transform, and Hilbert–Huang Transform, among others. Maciusowicz et al. [155] employed three time-frequency analysis methods (STFT, Continuous Wavelet Transform, and Smoothed Pseudo Wigner–Ville Transform) to process MBN signals, aiming to evaluate the magnetic anisotropy of electrical steel. Upon comparison, it was found that the STFT exhibited high sensitivity to the magnetization dynamics of the material and required the shortest computation time. Yang et al. [156] proposed a new MBN time-frequency characteristic parameter based on the Hilbert–Huang Transform, and they utilized this characteristic parameter to enhance the accuracy of stress detection in ferromagnetic materials. It should be noted that there is no standard answer for choosing which time-frequency analysis method to use for processing MBN signals. Specific situations require specific analysis.

Currently, key components may exhibit defects such as corrosion, mechanical damage, and material degradation when operating in complex environments. The occurrence of these defects often coincides with stress concentration and an uneven distribution of microstructure. MBN technology is also sensitive to this type of defect. Sagar et al. [157] studied the changes in MBN signals with early fatigue damage in low carbon steel, and they revealed multiple stages of early fatigue damage based on changes in dislocation density and microstructure. Jančula et al. [158] utilized MBN signal to monitor the corrosion degree of S460MC steel. Miesowicz et al. [159] used the signal fractal analysis method of Wavelet Transform to process MBN, extracting fractal parameters sensitive to fatigue microcracks, thus achieving the characterization of microcrack length. However, these parameters cannot be used for monitoring the initiation of microcracks. Blanco et al. [148] focused on the influence of crystal texture and microstructure on the early stage of crack formation and growth. From this perspective, there are numerous factors that affect the mechanical performance degradation of key components (microstructure, external load, etc.), and they exhibit strong coupling characteristics among each other. Therefore, it is required to extract multiple characteristic parameters and conduct multi-parameter fusion detection and analysis, constructing a multi-input, multi-output statistical and predictive model. This model aims to achieve the nondestructive evaluation of the degradation degree of mechanical properties of in-service structures, reducing the failure of key components.

## 5. Summary and Outlook

Magnetic anisotropy detection technology is a novel in situ nondestructive testing technique. With its nondestructive, rapid, efficient, and other unique advantages, it can achieve the detection and evaluation of stress states and textures in ferromagnetic materials, presenting significant application prospects. Since the 1990s, the research team of Queen’s University in Canada extensively explored magnetic anisotropy. At present, several countries including Germany, Mexico, Canada, Brazil, Poland and other countries have successively developed related devices and conducted extensive research, making significant progress in basic theory, experimental technology, and signal processing. However, the magnetic anisotropy was seldom reported, and the research findings were not abundant in China. Magnetic anisotropy detection is based on the theory of MBN anisotropy detection. Utilizing the sensitivity of MBN signals to the anisotropy of material structure and properties, the MBN measurement technology is extended to magnetic anisotropy detection, greatly expanding the application scope of this technology.

This paper investigates a large number of studies in the literature related to magnetic anisotropy detection, thoroughly combs its development process, and summarizes the main application progress of magnetic anisotropy detection technology. Although magnetic anisotropy detection technology is becoming increasingly mature, there are still some urgent issues to be addressed in theoretical research, experimental methods, signal processing, and engineering applications: 

(1) There is insufficient research on the evolution mechanism of MBN signals. At present, the generation mechanism of MBN has been basically understood both domestically and internationally, and some widely used MBN models have been developed. The magnetization mechanism considered in existing MBN models mainly is the irreversible motion of the domain wall. These models also cannot explain the anisotropic characteristics of materials. Therefore, it is impossible to provide accurate theoretical guidance for magnetic anisotropy detection, which is the fundamental aspect that restricts the development of magnetic anisotropy detection technology.

(2) The MBN experimental evaluation method for magnetic anisotropy needs to be improved. MBN signals are closely related to excitation magnetic field parameters, sensor structural dimensions, and other factors. There are relatively few studies in China evaluating the magnetic anisotropy of ferromagnetic materials based on MBN anisotropy. It is necessary to strengthen basic research on the effects of various detection parameters on the MBN anisotropy results so as to obtain a more comprehensive understanding of the influencing factors and the variation patterns of MBN signals, which lays a theoretical foundation for the application of magnetic anisotropy detection technology.

(3) The MBN signal processing method needs to be further explored. MBN anisotropy detection technology has been widely applied in stress detection and texture evaluation. However, the accuracy and precision of MBN anisotropy detection results are limited by the use of single time-domain or frequency-domain analysis methods. Therefore, it is necessary to introduce a new MBN signal processing method to specifically extract characteristic parameters containing key information about material states. The new method can enhance the reliability and accuracy of MBN anisotropy detection, which overcomes key technical challenges for the application of magnetic anisotropy detection technology.

(4) It is necessary to strengthen the research on the engineering application of magnetic anisotropy detection technology. The impact of environmental factors on MBN signals must be comprehensively considered during magnetic anisotropy detection. Factors such as temperature, electromagnetic interference, mechanical vibration, shock, and radiation continue to be significant constraints on the practical application of magnetic anisotropy detection technology in the field. In addition, there is currently no standardized MBN signal testing method, making it difficult for scholars to unify the relationship between MBN characteristic parameters obtained through experiments and mechanical properties. Therefore, targeted detection processes can be developed subsequently, and national standards for corresponding detection methods can be established, providing a basis for the promotion and application of magnetic anisotropy detection technology.

In this review, the authors have categorized the publications related to magnetic anisotropy into three topics depending on their content. Although various aspects of magnetic anisotropy have been discussed, some omissions may inevitably exist. Finally, the authors hope that this review can guide MBN evaluation and other related work on the magnetic anisotropy of ferromagnetic materials.

## Figures and Tables

**Figure 1 sensors-24-07587-f001:**
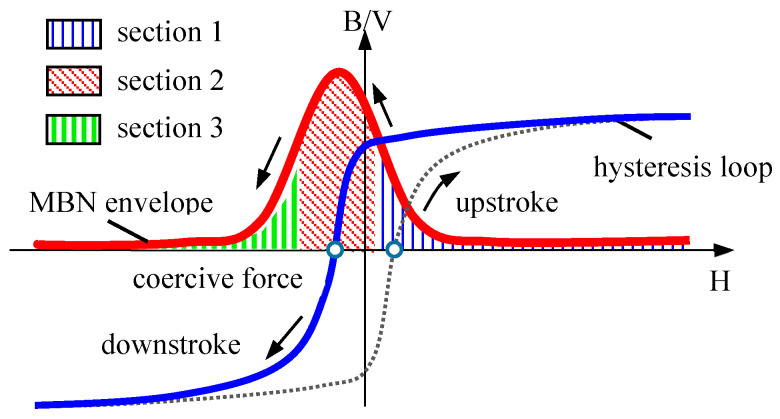
MBN envelope in different magnetization sections.

**Figure 2 sensors-24-07587-f002:**
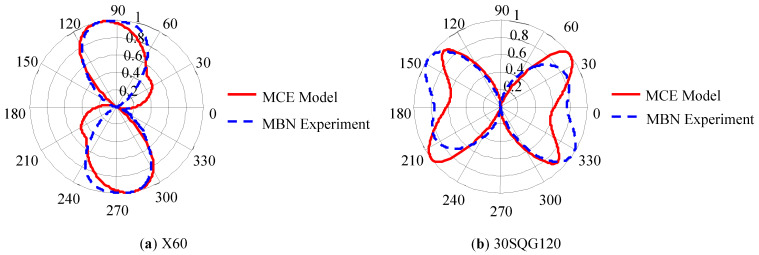
Normalized comparison between MCE model prediction results and MBN experimental results for typical materials.

**Figure 3 sensors-24-07587-f003:**
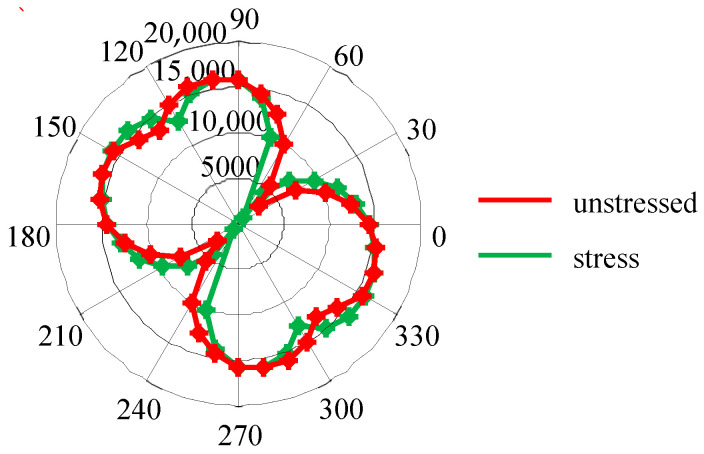
MCE simulation through modeling under the stress and unstressed.

**Figure 4 sensors-24-07587-f004:**
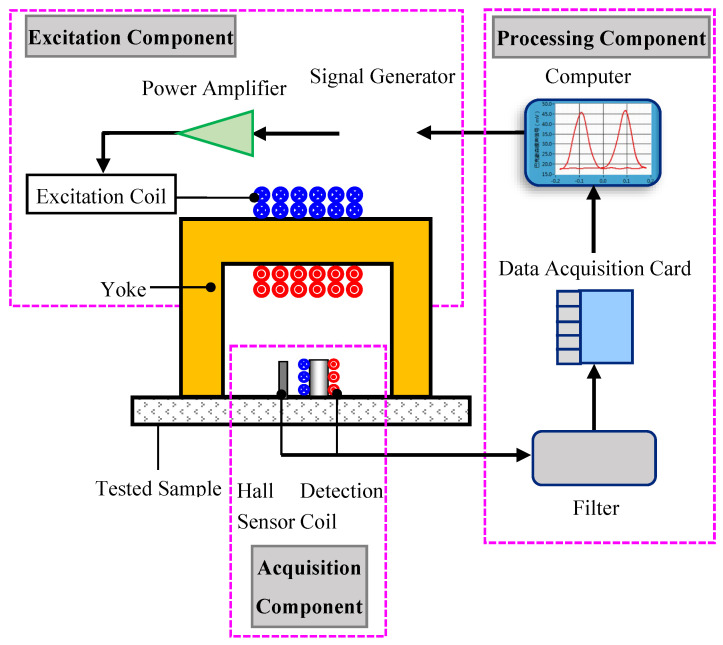
Schematic diagram of the MBN signal detection device.

**Figure 5 sensors-24-07587-f005:**
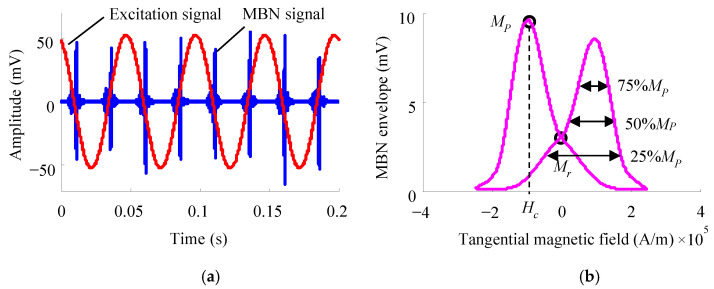
MBN signal: (**a**) MBN in time domain and (**b**) MBN butterfly curve.

**Figure 6 sensors-24-07587-f006:**
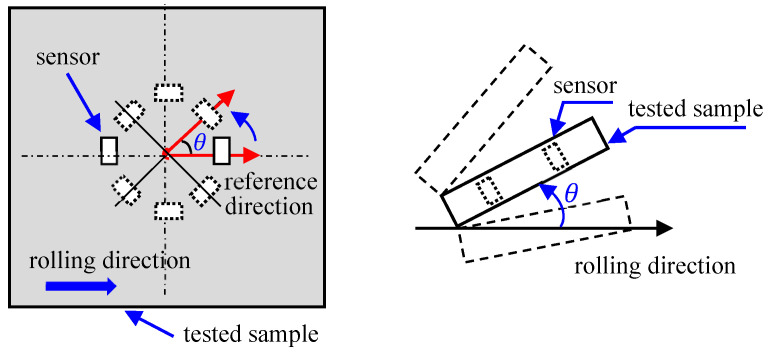
Schematic diagram of mechanical rotated MBN anisotropy detection device.

**Figure 7 sensors-24-07587-f007:**
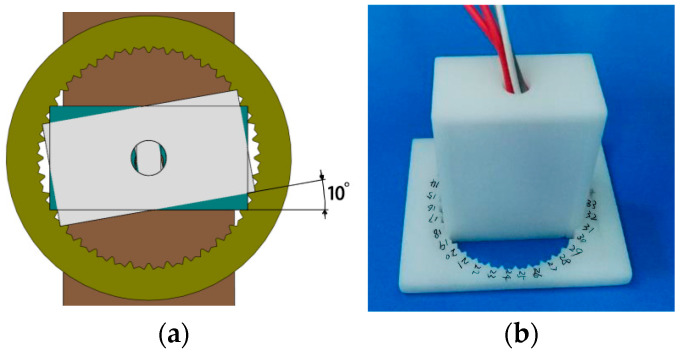
MBN anisotropy detection device with a fixation base: (**a**) Model diagram and (**b**) Physical picture.

**Figure 8 sensors-24-07587-f008:**
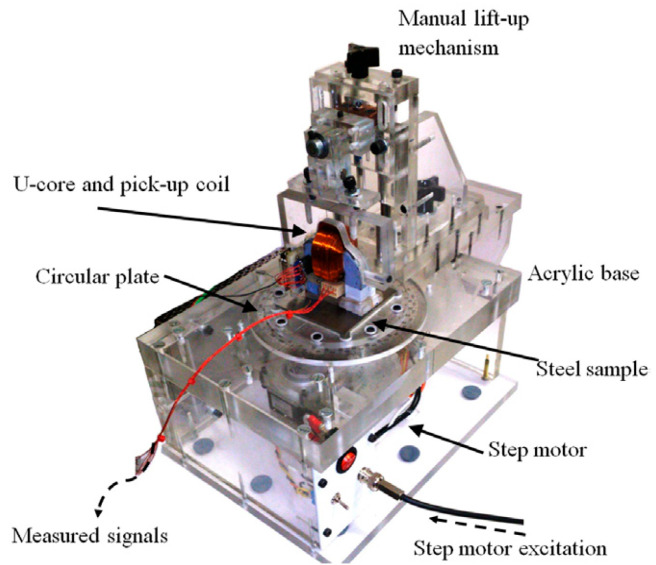
MBN anisotropy detection device with a goniometer.

**Figure 9 sensors-24-07587-f009:**
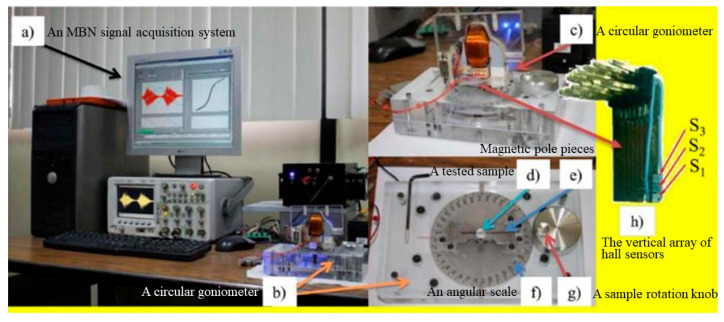
MBN anisotropy detection device for circular samples.

**Figure 10 sensors-24-07587-f010:**
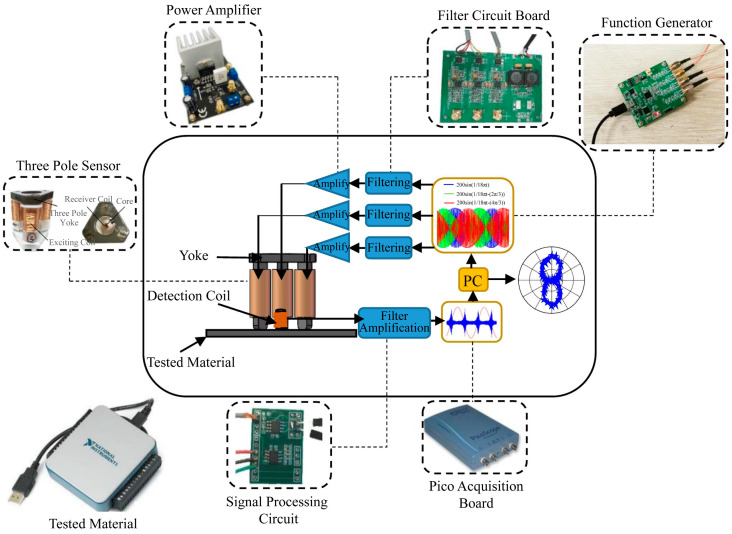
MBN anisotropy automatic detection system.

**Figure 11 sensors-24-07587-f011:**
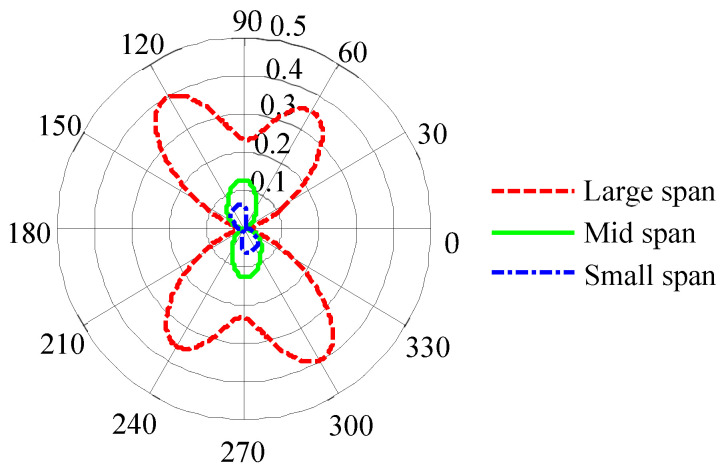
RMS pole figure measured under different spans.

**Table 1 sensors-24-07587-t001:** Comparison of MBN anisotropy detection devices.

Detecting Device	Detection Speed	Price	Detection Accuracy	Limitation	Scope of Application
Mechanical rotation	Slow	Cheap	High	The impact of human factors is significant	Plane component detection
Automatic rotation	Fast	Expensive	Higher	External factors interfere with MBN signals	Detection of plane and curved components
Multi-pole sensor	Fast	Expensive	Higher	It is necessary to ensure the uniformity of the central magnetic field	Online detection of ferromagnetic components

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
