# Peer review of "A Survey of the Magnetic Anisotropy Detection Technology of Ferromagnetic Materials Based on Magnetic Barkhausen Noise"

_sensors, 2024, doi:10.3390/s24237587_

Round 1
Reviewer 1 Report
Comments and Suggestions for Authors
The article is devoted to an important and relevant method of studying materials – the Bargauzen effect. The authors have constructed a review starting from the basics of the effect, touched upon the phenomenological description of the processes taking place, the technique of the experiment and modern works. The part related to the device of detecting anisotropy is well written.
The article is suitable for the subject of the journal – sensors, since the sensitivity of ferroprobes is determined by fluctuations in magnetic energy that occur during magnetization reversal. The stochastic nature of the emergence and destruction of domains during magnetization reversal leads to the fact that the magnetization dependence curve itself on the field applied to the magnet y consists of many steps associated with the processes of restructuring the domain structure – Bargauzen jumps.
Of the comments, it should be noted:
1)     Excessive citation of works by the same authors, sometimes the same type of works, for example, Krause T W -11 articles, Stupakov - 5 articles, He Y – 4 articles, Liu J - 4 articles, Maciusowicz – 3 articles and so on.
2)Â Â Â Â Â At the same time, informative and, one might say, fundamental works on this topic have been completely overlooked, for example,
Rudyak V.M., The Barkhausen effect, Soviet physics uspekhi, V.13, No.4. 1971 461-479. DOI: 10.1070/PU1971v013n04ABEH004681 or V.V. Volkov, V.A. Bokov, Domain wall dynamics in ferromagnets, Physics of the Solid State, Volume 50, pages 199–228, (2008). DOI: 10.1134/S1063783408020017
 Since, in my personal opinion, it is necessary to illuminate fundamental laws in an applied modern vision, the article can be published.
Reviewer 2 Report
Comments and Suggestions for Authors
Magnetic anisotropy detection technology using Magnetic Barkhausen Noise (MBN) method, as a nondestructive testing approach, indeed represents a transformative advance in material science and engineering due to its capacity for rapid, efficient, and precise evaluation of stress states and textures in ferromagnetic materials, without causing any structural damage to the materials under study. As authors discussed here, this method has been extensively explored since the 1990s by different international research teams, have invested significantly in research and development to refine the foundational theories, experimental techniques, and signal processing frameworks associated with magnetic anisotropy detection, leading to important advancements.
a) How does magnetic anisotropy in ferromagnetic materials impact MBN signal characteristics, and what are the specific features of MBN signals that correlate with different levels or types of anisotropy?
b) In this manuscript, effect of different factors such as sensor design, magnetic excitation parameters) on MBN signal sensitivity to stress and strain variations in ferromagnetic materials is not discussed properly. How can these parameters be optimized for accurate stress detection?Â
c) Please include a comparative table on MBN noise amplitudes based on different sensors data (from literatures).Â
d) How can MBN signal processing techniques be enhanced to improve the detection of subtle variations in material properties, and which combination of time-domain and frequency-domain analyses yields the most accurate material state information?
e) Please discuss role of environmental effects (such as temperature changes, mechanical vibrations, and electromagnetic interference) on MBN signal stability and accuracy.
f) Can MBN techniques be extended to detect complex internal stresses or material degradation in 3D configurations, and what advancements in sensor technology are needed to achieve such applications? Please add this part with a short discussion.
  Â
Minor comments:Â
g) Please rewrite the abstract and make it more concise.
h) Kindly change the caption of Figure 5. If MBN noise is random then how you referred as typical.
i) If you have two sub-figures in a figure, you must index it and mention it. As an example, from Fig. 7 anything is not clear.
j) I did not find any reference of this cited paper. Please add doi. Wang X. Multi-angle magnetic Barkhausen noise detection method and system for residual plastic deformation of carbon steel[D]. Beijing University of Technology, 2020. Same for ref. [99], [102]
k) Please include the following references which is relevant and missing in this manuscript.Â
a) ‘Operational Parameters for Sub-NanoTesla Field Resolution of PHMR Sensors in Harsh Environments. Sensors 2021, 21, 6891.’
b) Magnetic, magnetoresistive, and low-frequency noise properties of tunnel magnetoresistance sensor devices with amorphous CoFeBTa soft magnetic layers. J. Phys. D Appl. Phys. 2021, 54, 095002.
Â
Â
Reviewer 3 Report
Comments and Suggestions for Authors
The paper provides a review of the technique for assessing the direction of the easy magnetization axis in steel products based on the measurement of Barkhausen noise. Although the Barkhausen noise measurement method has a very deep history, magnetic anisotropy assessments using this method have been developed relatively recently. The review provides general ideas about the method, considers models for the origin of anisotropy, equipment used for measurements, and provides comparisons of measurements and models. In general, the review makes a good impression: it is easy and interesting to read. It is evident that the authors have worked through a very solid layer of literature. I believe that such a review is well suited to the subject of the journal; it can really serve as a guide for assessing the quality of steel products and magnetic flaw detection. The article can be accepted after the authors have worked through the comments below.
Â
1. It would be interesting to see the following discussion in the text. The methods of MBN measurements assume that the signal comes from some depth of the material. If the sample is very thick, the information will only come from the near-surface region. Which one? Presumably, this will determine the size of the yoke? Does this mean that a thicker steel wall will require a larger yoke?
2. In magnetic flaw detection, it is interesting to study the anisotropy associated with structural defects at different scales. What are the capabilities of this technique for assessing extended and localized structural defects?
3. What is shown in Fig. 2? This is not clear from the caption or the text. What value is plotted along the radius? How was it measured? If RMS is meant (explanations in Fig. 5), then this is not enough. Explanations are needed for Fig. 2 at the moment it is looked at. I do not see equations for the RMS value in the text.
4. How were the data in the model calculated in Fig. 2? Can any of the equations be referenced in the text? If the data are from another paper, a reference is required.
5. The caption to Fig. 3 should either reference the equations or the paper.
6. It is good that the authors provide equations to explain the models. However, the meaning of the symbols should be explained. For example, equation (6). What is E_NMB?
7. The text gives many equations with explanations. But in my opinion, there is also a lot of confusion. For example, in equation (1), t is time, and in equation (4), t is angle. It is better to give the angle in equation (1) using an over symbol.
8. Lines 802-804 say "Finally, the author hopes that this review can guide MBN evaluation…". Does only one author express this hope? Which one?
9. Lines 195-197 say "This model has also been recognized by domestic and international research teams …". Mentioning domestic teams seems inappropriate for a general reader.
10. Equation (1) is missing a subscript.
11. Line 155 says "Eq. (1) is a step function." This should be rewritten. An equation and a function are different concepts.
Round 2
Reviewer 2 Report
Comments and Suggestions for Authors
In the revised manuscript version, the MBN anisotropy model is much more structured
and concise. Please review the minor comments, which might be included in the revised
version.
1.     In the … easy magnetization is <100>. Is it always true?
2.     In polycrystalline materials… magnetic anisotropy. Is compressive stress involved in the MBN effect of polycrystalline material?
Â
3.     ‘What are the specific features of MBN signals that correlate with different levels or types of anisotropy?’ Please include your comments with figures in the main text.
